# Research of ZnO Arrester Deterioration Mechanism Based on Electrical Performance and Micro Material Test



**Qizhe Zhang [1,\*], Shenghui Wang [1], Xinghao Dong [2], Mingliang Liu [1], Qi Ou [1] and Fangcheng Lv [1]**

[1] State Key Laboratory of Alternate Electrical Power System with Renewable Energy Sources, North China Electric Power University, Beijing 102206, China; hdwsh@ncepu.edu.cn (S.W.); 120192201378@ncepu.edu.cn (M.L.); 120202201103@ncepu.edu.cn (Q.O.); lfc@ncepu.edu.cn (F.L.)

[2] Hebei Provincial Key Laboratory of Power Transmission Equipment Security Defense, North China Electric Power University, Baoding 071003, China; 1182101037@ncepu.edu.cn

[\*] Correspondence: sxjczqz@163.com; Tel.: +86-156-0046-2500

**Abstract:** The traction power supply system of an Electrical Multiple Unit (EMU) often suffers from overvoltage impact. As an important protection device for on-board electrical equipment, the working environment of a roof arrester is worse than that of a power system. In recent years, the explosion failure of the roof arresters of an EMU has occurred from time to time, which seriously endangers the safe operation of high-speed railways. In this paper, the electrical performance test and material micro test of roof arrester in three states of normal, defect, and exploded, are carried out in order to study the internal causes of roof arrester explosion and clarify its deterioration mechanism. Using the DC reference voltage test and leakage current test, the electrical performance differences of normal, defective, and exploded arresters are obtained. By studying the disassembly of an arrester, the appearance characteristics of arrester varistor in three states are obtained. The micro morphology and chemical elements of the varistor are analyzed by Scanning Electron Microscope and Energy Dispersive Spectrometer. The deterioration mechanism of the arrester varistor is then revealed, and preventive measures for the explosion failure of the roof arrester are put forward. The obtained results show that, during the long-term operation of the roof arrester of an EMU, the varistor may be damp, and therefore the aluminum electrode layer and side insulation layer of the varistor may deteriorate. After the deterioration of the aluminum electrode layer, the content of the O element increases, and multiple film structures are formed on the surface. After the deterioration of the side insulating layer, the content of the O element increases, and the surface becomes uneven. Improving the sealing performance requirements of the roof arrester and optimizing the maintenance process can reduce its explosion failure.

**Keywords:** EMU; arrester; explosion; microscope test

## 1. Introduction

An lightning arrester can effectively limit the amplitude of overvoltage and protect the safety of back-end electrical equipment [1]. During operation, the arrester may withstand an overvoltage impact, high salt fog environment, surface pollution, moisture, and other complex environments, and therefore it may fail. The common faults of a lightning arrester include bursting of the varistor, bursting of the insulating cylinder, and external pollution flashover. Quality problems or serious aging of the arrester accounts for a large proportion of the total number of arrester faults [2]. A locomotive arrester is mainly composed of a silicone rubber composite jacket, core, wiring terminal, and flange [3]. The varistor is the core component of a metal oxide arrester [4]. The characteristics of the varistor directly define the electrical performance of the arrester. The features of bulk varistors are influenced by the geometry and the topology of the granular microstructure, as well as the properties and the distribution of the electrical characteristics of grain boundaries [5]. The main technically and scientifically challenging topics of varistors are the filamentary current flow

in real microstructures, the inhomogeneities present on all length scales, and the long-term stability [6]. The electrical characteristics of a zinc oxide (ZnO) varistor are determined by its microstructure [7]. Any non-uniformity will lead to the reduction of the varistor characteristics [8]. The composition and manufacturing process factors of the varistor directly affect its different required properties [9]. Therefore, the performance deterioration of the arrester is closely related to the change of the varistor micro characteristics.

Over the past decades, several studies have been carried out in the fields of lightning arrester fault mechanisms [10], aging characteristics, and fault detection. It is a well-established fact that varistor failures, namely, pinhole and cracking, can be attributed to varistor structural and electrical nonuniformity. Pinhole failure usually occurs at TOV conditions, while cracking occurs during surge currents [11]. The authors in [12] proposed that the failure mechanism of a ZnO varistor, under the action of 2 ms square wave, includes a pinhole failure mode and a burst mode. The analysis of arresters requires transient and coupled finite element simulation of the mutually-dependent electric and thermal fields, employing an accurate electrothermal model [13]. The authors in [14] explore the failure mode of ZnO varistors under multiple lightning strokes. The authors in [15] study the electrical-thermal multiple-field analysis on the degradation and failure of an 110 kV-class surge arrester subjected to successive lightning current impulses by experiment and simulation. Xiaochuan [16] studied the micro morphology of the square wave burst varistor. He pointed out the shortcomings of the burst failure mode and demonstrated that the ZnO vapor shock wave, generated by arc combustion in pinholes, is the cause of varistor burst. The authors in [17] studied the overheating issue of the arrester under Ultra Harmonics Overvoltage. Zhipeng [18] carried out an artificial accelerated aging test on a 110 kV MOA varistor. He showed that high temperature and moisture will reduce the insulation resistance of the varistor. He also studied the voltage distribution and heating of the whole arrester in the presence of a degraded varistor through simulation calculation. Chubiao et al. [19] studied the lightning impulse aging characteristics of a 110 kV whole zinc oxide arrester. They obtained the changes of a DC reference voltage and residual voltage with impulse times. The aged arrester was disassembled and inspected, the surface flashover trace of the varistor was observed, and the reason for the aging of the arrester under impulse current was explained using micro test research. Tong et al. analyzed and summarized the common fault types and characteristics of MOA. They also analyzed the advantages of the leakage current and insulation resistance tests [20]. In [3], the authors disassembled a cracked locomotive lightning arrester and found that there were traces of moisture in the metal parts and in the varistor inside the lightning arrester.

Lightning arrester failure is mainly caused by internal defects and external overvoltage. The main internal defects are damp and the deterioration of the varistor. The causes of damp or water ingress of the varistor may include poor sealing, an unqualified assembly environment, and damage due to external forces. When the arrester is damp or deteriorated, its full current and resistive current may change. Therefore, a defective arrester can be detected using a leakage current test [21,22]. Several monitoring devices were developed for the arrester on-line test [23–25]. Based on the test results, state detection methods were proposed [26]. Besides the internal defects of the arrester, external overvoltage is also an important cause of arrester failure. When the overvoltage amplitude and duration are large enough, the intact arrester may also break down [27]. The ability of the lightning arrester to withstand overvoltage can be improved by setting parallel clearance [28]. A lightning arrester with a gap for catenary for a high-speed railway has been successfully developed in [29].

Although the gap arrester can withstand an increased energy overvoltage impact, the active EMU roof arrester does not have a gap. In addition, the overvoltage of the EMU traction power supply system is more frequent than that of the power system. The EMU roof arrester may act more frequently, and a burst fault occurs from time to time. Because the voltage monitoring device of an EMU can only record the effective value of voltage in a certain period, it cannot obtain the voltage and current waveform data at

the time of lightning arrester failure. The reason for explosion can only be analyzed by disassembling the fault lightning arrester. Studies have shown that the electrical performance of the deteriorated arrester will decline. However, the relationship between the electrical performance and the varistor physical and chemical characteristics should be further studied.

In this paper, three samples of a normal, defective, and exploded arrester were selected to carry out experimental research of the roof arrester of an EMU. The overall electrical performance of the arrester was obtained by DC reference voltage tests and leakage current tests. The arresters were then disassembled and appearance inspection and electrical performance tests for arresters in different states were conducted. The microscopic characteristics of the arrester varistors under different states were obtained by SEM and EDS. The possible causes of lightning arrester explosion were put forward through analysis.

## 2. Test Object and Research Method

### 2.1. Test Object

In this paper, the used test object is the roof arrester for an EMU. The performance parameters of the arrester are:

(1)     Rated voltage: 42 kV;
(2)     Nominal discharge current: 10 kA;
(3)     Lightning impulse residual voltage under nominal discharge current: <105 kV;
(4)     DC1mA reference voltage: >58 kV;
(5)     Continuous operating voltage: 34 kV;
(6)     Capacitance of each varistor: 530 pF;
(7)     2 ms square wave current capacity: 500 A.

Note that the "normal" lightning arrester test object is a new lightning arrester that is not in service. The "defective" arrester test object is an unqualified arrester detected in routine maintenance. The test object of "exploded" arrester is an arrester with burst fault during service. The ex-factory performance parameters of a normal arrester and a defective arrester are shown in Table 1.

**Table 1.** Factory performance parameters of a roof arrester.

| Status | $U_{1mA}$ /kV | Idc@0.75$U_{1mA}$ /µA | $U_r$ /kV | $I_X$/µA | $I_R$/µA |
|---|---|---|---|---|---|
| Normal | 60.6 | 7.0 | 104.4 | 485.0 | 90.0 |
| Defective | 62.3 | 7.5 | 104.1 | 498.0 | 105.0 |

In Table 1, U1mA is the DC reference voltage, I0.75u1ma is the leakage current under the DC reference voltage of 0.75 times rated, Ur is the lightning impulse residual voltage under the nominal discharge current, and IX and IR are the full current and resistive current under the continuous operation voltage (34 kV), respectively.

The factory test data of the burst arrester is unknown. It is important to mention that the weather was fine when the fault occurred, the train did not operate the pantograph (a lifting device to realize the connection between train and catenary), the circuit breaker was not opened or closed, and the overvoltage waveform was not recorded. A photo of the pressure relief part of the burst arrester test object is shown in Figure 1.

### 2.2. Research Methodology

As the burst arrester has been broken down, the electrical performance test of the whole arrester cannot be carried out. Firstly, the overall performance test of the normal and defective arresters is carried out. The test items include DC1mA reference voltage, leakage current under 0.75 times DC reference voltage, and full current and resistive current tests under a continuous operation voltage. The ambient temperature at the time of the test is 25 °C. Data are measured after 1 s to avoid transient phenomena.

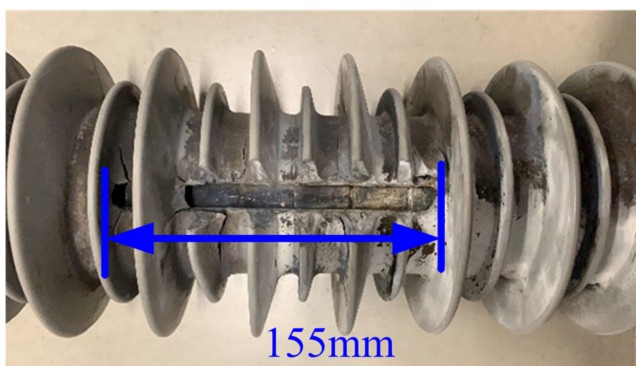

**Figure 1.** Exploded arrester.

The test circuit diagram of the leakage current, under DC reference voltage and 0.75 times DC reference voltage, is shown in Figure 2.

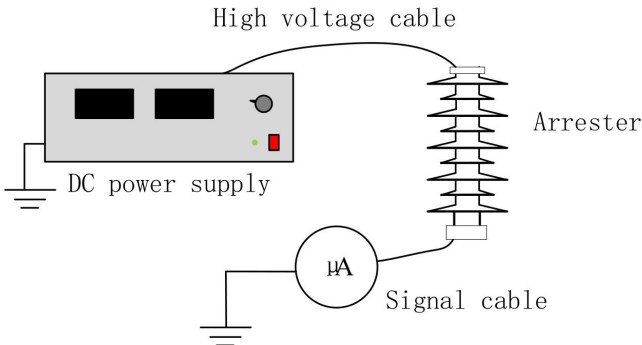

**Figure 2.** DC Voltage test circuit.

A DW-P104-2ACF2 DC high voltage power supply is selected, with an output voltage range of 0-100kV, time drift accuracy of 0.1%/h, and temperature drift accuracy of 0.1/°C. A DC high voltage is first applied to the arrester. The voltage is then gradually boosted until the current flowing through the arrester reaches 1 mA. Afterwards, the DC reference voltage of the arrester is recorded. Finally, the voltage is reduced to 0.75 times the DC reference voltage, and the corresponding leakage current is recorded. The full current and resistive current test circuit, under a continuous operation voltage, is shown in Figure 3.

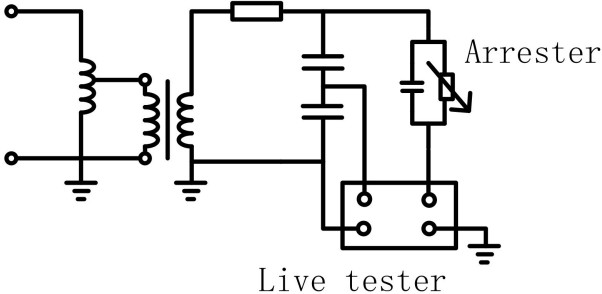

**Figure 3.** Leakage current test circuit under continuous operation voltage.

In the power frequency test transformer, the rated voltage at the high voltage side is 110 kV, the rated capacity is 10 kVA, and the transformation ratio of the capacitor voltage divider is 1000:1. An AI-6103 zinc oxide arrester live tester is used to measure the leakage current of the arrester. A 34 kV continuous operation voltage is applied to the arrester, then the full current and resistive current of the arrester are recorded.

The arrester is disassembled after the electrical performance test is completed. A visual inspection and electrical performance test are conducted on the disassembled varistor. The electrical performance test principle is the same as that of the whole arrester. Finally, SEM and EDS tests are carried out on the varistor of the arrester in order to analyze the deterioration mechanism of the arrester, from the perspective of micro morphology and chemical composition.

### 3. Macroscopic Characteristics of Roof Arrester in Different States

*3.1. Overall Electrical Performance*

The DC voltage test and the continuous operation voltage test results of normal and defective arresters are shown in Table 2.

**Table 2.** Overall electrical performance of roof arrester.

| | Status | $U_{1mA}$/kV | $I_{0.75U1mA}$/µA | $I_X$/µA | $I_R$/µA |
|---|---|---|---|---|---|
| **Normal** | Factory value | 60.6 | 7.0 | 485.0 | 90.0 |
| | Test value | 60.7 | 8.7 | 525.0 | 102.8 |
| | Rate/% | 0.2 | 24.3 | 8.2 | 14.2 |
| **Defective** | Factory value | 62.3 | 7.5 | 498.0 | 105.0 |
| | Test value | 58.3 | 157.0 | 576.0 | 257.0 |
| | Rate/% | 6.4 | 1993.3 | 15.7 | 144.8 |

It can be seen from Table 2 that, compared with the factory test value, the defective arrester test sample has the phenomena of DC reference voltage decrease, leakage current increase under 0.75 times DC reference voltage, and resistive current increase under continuous operation voltage. The DC reference voltage is close to 58 kV (the lower limit specified in the IEC standard), and the change rate is 6.4%. The leakage current at 0.75 times the DC reference voltage is much higher than 50 µA (the upper limit value specified in the IEC standard), and the change rate reaches 1993.3%. The change rate of resistive current of the defective arrester is 144.8%, which is much higher than that of the normal arrester by 14.2%. The test value of the normal arrester is close to the factory value.

*3.2. Appearance Characteristics of Varistors*

In this section, the three arrester samples are disassembled and the characteristics of each arrester varistor are observed. The typical appearance of a normal arrester varistor is shown in Figure 4.

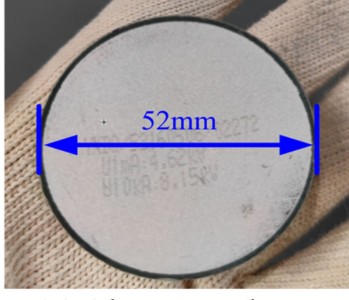

(**a**) Aluminum layer

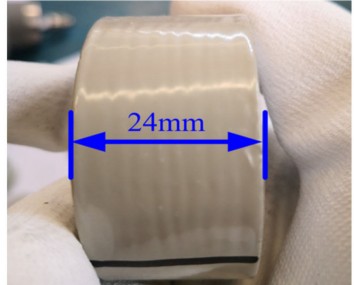

(**b**) Side insulation

**Figure 4.** Typical appearance of a normal arrester varistor.

The surface of each normal arrester varistor is flat and clean, and it is free of abnormalities. The typical appearance of a defective arrester varistor is shown in Figure 5.

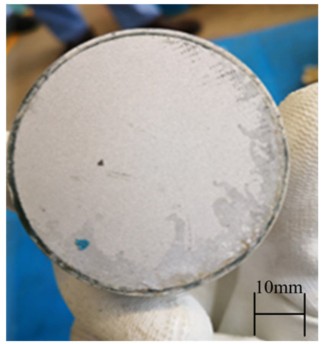

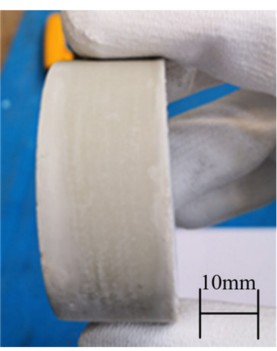

(**a**) Aluminum layer          (**b**) Side insulation

**Figure 5.** Typical appearance of a defective arrester varistor.

It can be seen that the varistor appearance near the high voltage end of the defective arrester is different from that of the normal arrester. Suspected chemical corrosion traces appeared at the edge of the aluminum electrode layer. The roughness of the side insulating layer of the varistor increases, and the color unevenness of the insulating layer also increases. The appearance of the burst arrester varistor is shown in Figure 6.

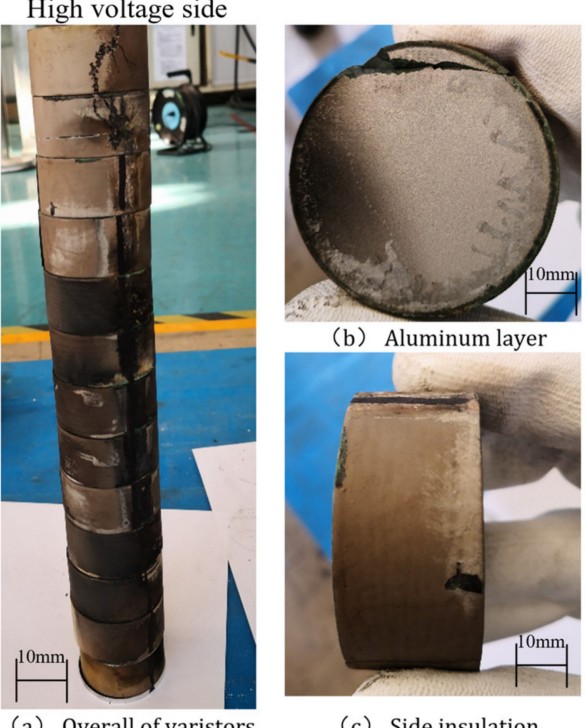

**Figure 6.** Typical appearance of a defective arrester varistor.

It can be seen from Figure 6a that the side insulating layer flashover first occurs on the varistor near the high-voltage end of the arrester. When the number of side flashover varistors reaches four, the remaining varistors bear all the voltage at both ends of the arrester, exceeding the varistor thermal limit. This results in a thermal collapse of the varistor. It can also be seen that there are suspected chemical corrosion traces on the edge of the aluminum electrode layer of the varistor. This is similar to the appearance of the aluminum electrode layer of the defective arrester. In addition, the morphology of

the insulating layer on the side of the varistor is similar to that of the defective varistor, while the color of the insulating layer is inconsistent.

In conclusion, the suspected corrosion traces at the edge of the aluminum electrode layer, and the morphological changes of the side insulation layer, may be the internal causes of the arrester failure. The EMU can timely disconnect the circuit breaker and lower the pantograph when the power frequency voltage is greater than 30 kV, but the EMU cannot monitor the impulse current and impulse voltage and the train often suffers from overvoltage during operation. The EMU cannot operate the circuit breaker and the pantograph to avoid overvoltage attack, and can only absorb overvoltage energy through the roof arrester. There are several methods in the surge protection industry to detect and disconnect defective arresters [30–33]. It is necessary to carry out relevant research to realize the on-line monitoring of roof arresters and reduce the occurrence of explosion accidents.

### 3.3. Electrical Performance of Varistors

In order to further study the influence of the appearance change of a defective arrester varistor on the electrical performance, an electrical performance test on the defective varistor is carried out.

A V-I test device is used to test the lightning impulse residual voltage under DC reference voltage and nominal discharge current. Factory test data are printed on the surface of each arrester varistor. The factory value and test value of each varistor are shown in Table 3.

**Table 3.** Electrical performance test of varistors.

| No. | $U_{1mA}$/kV | | | $U_r$/kV | | |
|---|---|---|---|---|---|---|
| | Factory Value | Test Value | Rate/% | Factory Value | Test Value | Rate/% |
| 1 | 5.2 | 5.1 | 1.5 | 8.7 | 8.7 | 0.4 |
| 2 | 5.4 | 5.3 | 0.7 | 9.0 | 9.0 | 0.8 |
| 3 | 4.9 | 4.9 | 0.6 | 8.4 | 8.4 | 0.1 |
| 4 | 5.2 | 5.2 | 1.2 | 8.7 | 8.7 | 0.8 |
| 5 | 5.4 | 5.3 | 1.3 | 8.9 | 8.8 | 0.3 |
| 6 | 4.9 | 4.9 | 0.8 | 8.4 | 8.4 | 0.2 |
| 7 | 5.4 | s5.3 | 1.1 | 8.9 | 8.9 | 0.1 |
| 8 | 5.2 | 5.2 | 1.0 | 8.7 | 8.7 | 0.0 |
| 9 | 5.2 | 5.2 | 0.8 | 8.7 | 8.7 | 0.3 |
| 10 | 5.2 | 5.2 | 0.4 | 8.7 | 8.8 | 0.7 |
| 11 | 5.4 | 5.3 | 1.1 | 8.9 | 8.9 | 0.5 |
| 12 | 5.2 | 5.2 | 0.8 | 9.0 | 8.8 | 2.5 |
| Total | 62.6 | 62.1 | 0.8 | 104.7 | 104.7 | 0.1 |

It can be seen from Table 3 that a small difference between the electrical performance test results of each defective arrester varistor and the factory test results exists. The DC reference voltage sum of each varistor is 62.1 kV, which is quite different from the one measured by the whole arrester (58.3 kV). This difference is due to the fact that, when the arrester is not disassembled, there is water vapor inside and the latter cannot be discharged, resulting in the wet state of the varistor, which shows that the DC reference voltage test value of the whole arrester is low. After disassembly, the arrester varistor changes from a wet state to a dry state in a ventilated and dry environment, which demonstrates that the electrical performance test results of the varistor are close to the factory values. Consequently, it can be deduced that the electrical performance test of the varistor after the disassembly of the arrester cannot truly reflect the defects of the varistor. In addition, the electrical performance test of the whole arrester can easily detect the internal defects. Simultaneously, if the maintenance interval of the arrester is too large, the damp arrester may return to dryness, resulting in internal defects that cannot be detected in time. Therefore, the arrester operates

with faults. When the defective arrester gets damp again, the safety of the EMU electrical equipment will be seriously threatened.

The lightning impulse residual voltage test results, under DC reference voltage and nominal discharge current of each defective arrester varistor, do not significantly change, compared with the factory value. Therefore, in order to study the relationship between the defect of the arrester varistor and the electrical performance, a square wave impulse current test and a high current impulse withstand test are carried out on the defective varistor. Two defective arrester varistors (numbered 1 and 2) are selected to carry out the 2 ms square wave impulse current test. The amplitude of the 2 ms-square wave current is 500A (i.e., the current capacity of the arrester), while the number of shocks is 20. The varistor passed the square wave impulse current test without damage. Two defective arrester varistors (numbered 3 and 4) are then selected for the high current impulse withstand test. Afterwards, 4/10 µs of 100kA amplitude is applied twice to each varistor.The varistor should be fully cooled between the two current shocks. If the varistor does not break down after two high current shocks, then it passes the test. The thickness of the varistor blocks is 24 mm. The residual voltage at 100 kA is 10.1 kV. The electric field stress (residual voltage peak/thickness) is 0.42 kV/mm. The current withstand test results of the defective varistor are shown in Table 4.

**Table 4.** Current withstand test of a defective varistor.

| No. | Test Program | Breakdown (Y/N) | Passed (Y/N) |
|---|---|---|---|
| 1 | 2 ms square impulse | N | Y |
| 2 | 2 ms square impulse | N | Y |
| 3 | First high current shock | Y | N |
|  | Second high current shock | / |  |
| 4 | First high current shock | N | N |
|  | Second high current shock | Y |  |

It can be seen from Table 4 that both defective varistors have passed the 2 ms square wave test. This indicates that the defective valve piece can meet the flow capacity requirements of the arrester when subjected to a continuous high current impact. Varistor No. 3 had a surface breakdown in the first high current impulse test, while varistor No. 4 had a surface breakdown in the second high current impulse test, and therefore it failed to pass the high current withstand test. To sum up, the varistor defect of the arrester is mainly reflected in the decline of the side insulation performance. When the arrester is impacted by a large current, the defective varistor is more prone to surface breakdown of the side insulation layer. The surface breakdown trace of the varistor in the high current impact test is shown in Figure 7.

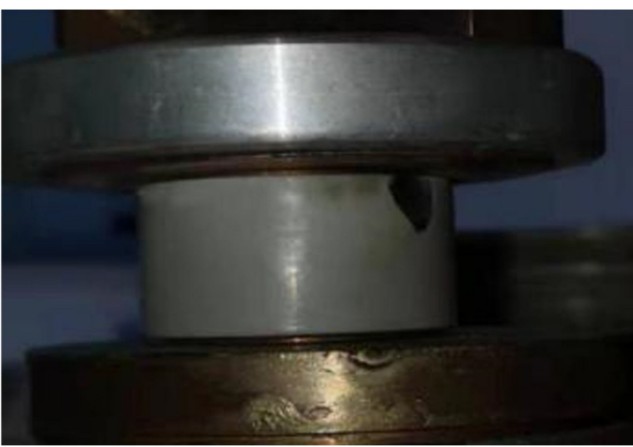

**Figure 7.** Surface breakdown of a varistor under a high current impact.

## 4. Deterioration Mechanism of Varistors

*4.1. SEM Test of Varistors*

There are significant differences in the appearance of arrester varistors in normal, defective, and cracked states. In order to study the relationship between the macro and micro characteristics of varistors, the micro morphology of an aluminum electrode layer and side insulation layer of varistors in different states were tested using Scanning Electron Microscope (SEM). An Energy Dispersive Spectrometer (EDS) analyzer is used to analyze the chemical elements of the aluminum electrode layer and side insulation layer of the varistor, in different states. The SEM test results of the varistor samples in different states are shown in Figure 8, with a magnification of 2K.

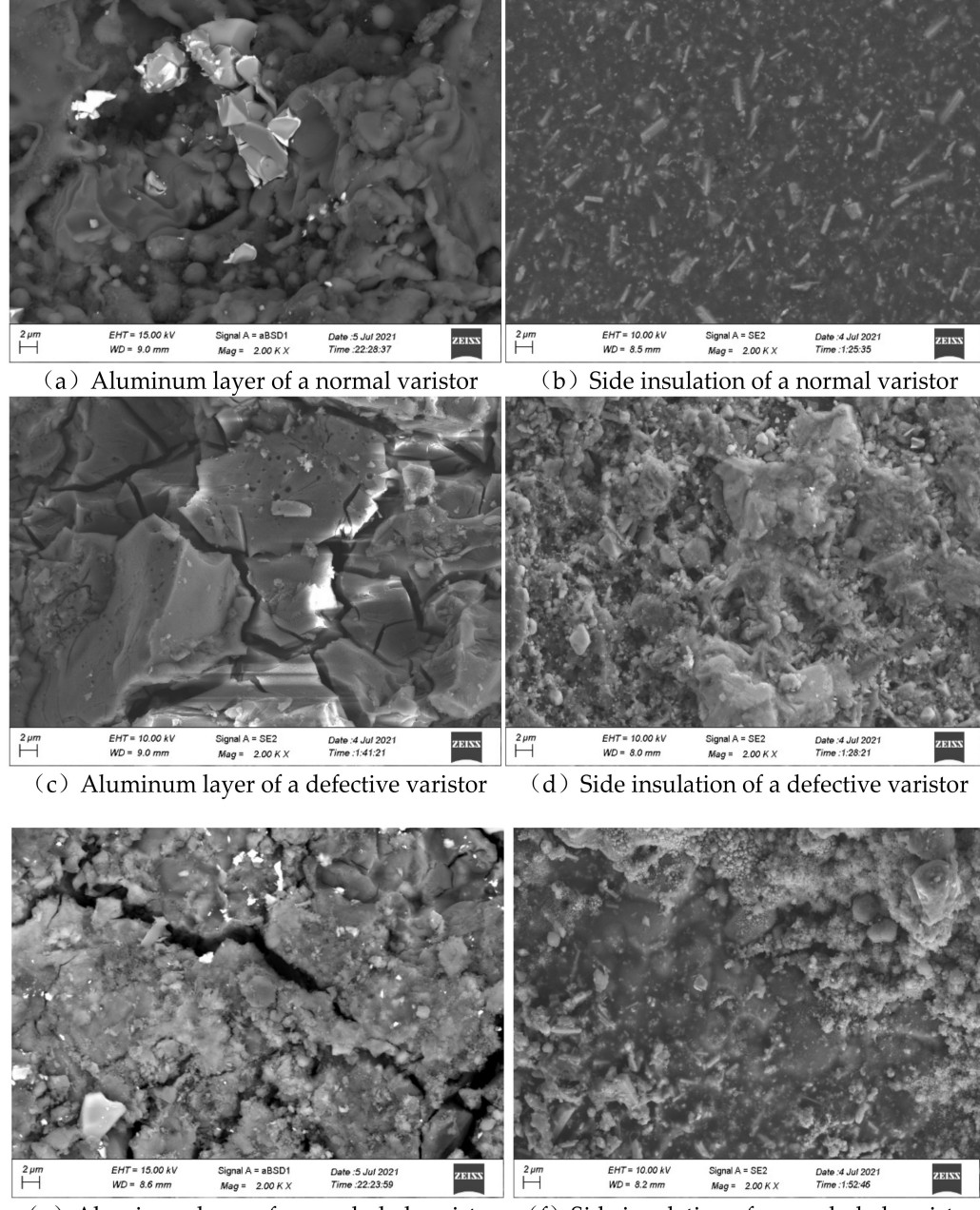

（a）Aluminum layer of a normal varistor　　（b）Side insulation of a normal varistor

（c）Aluminum layer of a defective varistor　　（d）Side insulation of a defective varistor

（e）Aluminum layer of an exploded varistor　　（f）Side insulation of an exploded varistor

**Figure 8.** SEM test of arrester varistors.

It can be seen from Figure 8 that there are great differences in the micro morphology of the aluminum electrode layer of the arrester varistor in the three states of normal, defect,

and burst. The surface morphology of the aluminum electrode layer of the normal varistor is flat. A plurality of film structures is formed on the surface of the aluminum electrode layer of the defective varistor. In addition, the scale of large films can reach 10 μm. A gully with a width of 1 μm exists between the films. Compared with the defective varistor, the film size of the aluminum electrode layer of the burst varistor is highly increased. The width and length of the gully between the films are also increased. The insulating layer on the side of the normal varistor is flat, while the strip packing is evenly distributed. The side insulating layer of the defective and cracked arrester varistor has no visible strip filler, and the morphology is uneven. Finally, the formation of a thin film of aluminum electrode layer and the roughness of the side insulation layer may be the reasons for the deterioration of the arrester varistor performance. The reasons for the difference in the micro morphology of the aluminum electrode layer of the varistors in different states will be discussed in Section 4.2.

### 4.2. EDS Test of Varistors

The EDS spectrum analysis results of each varistor are shown in Figure 9.

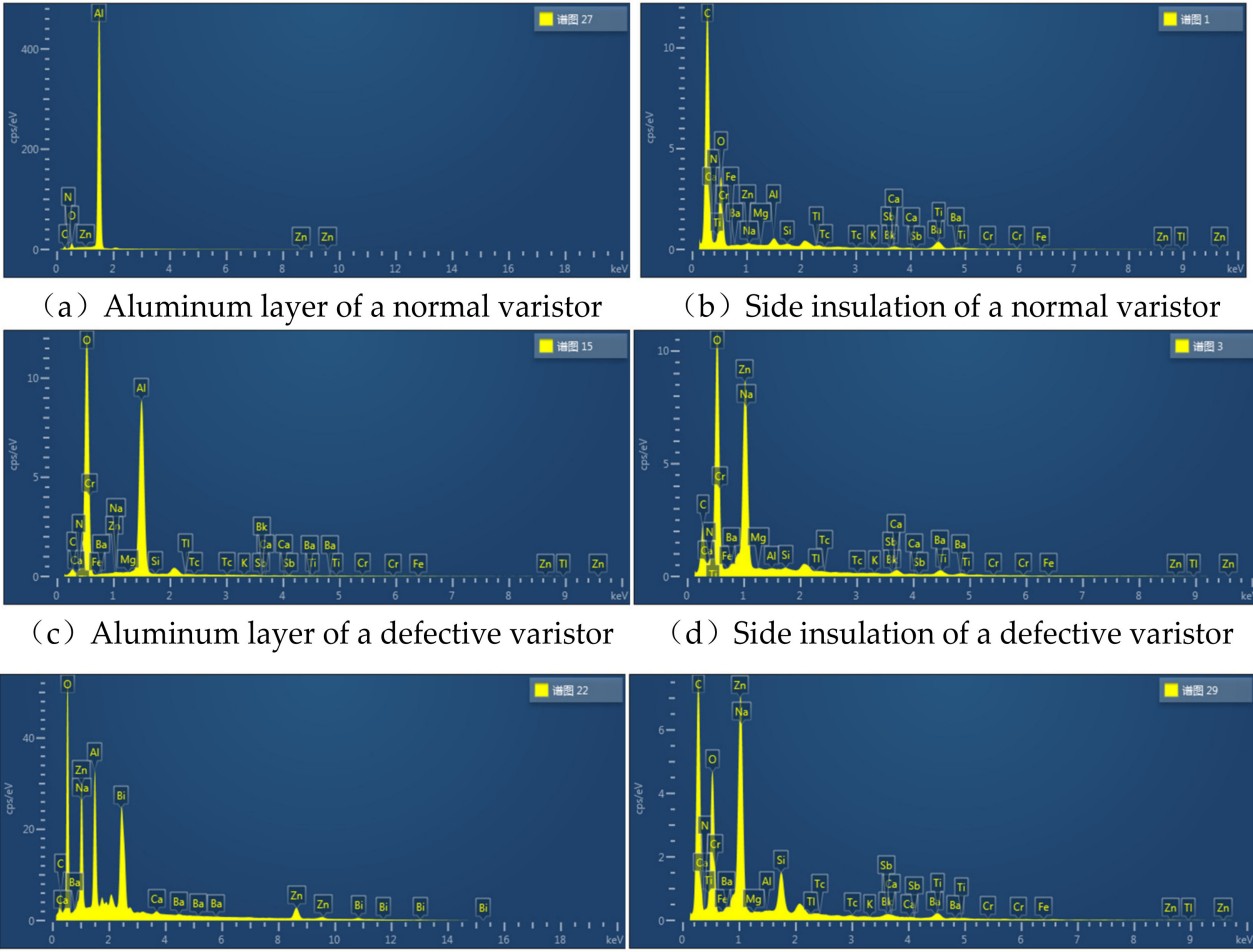

（a）Aluminum layer of a normal varistor　　　（b）Side insulation of a normal varistor

（c）Aluminum layer of a defective varistor　　　（d）Side insulation of a defective varistor

（e）Aluminum layer of an exploded varistor　　　（f）Side insulation of an exploded varistor

**Figure 9.** EDS test of arrester varistors.

It can be seen from Figure 9 that the main element of the aluminum electrode layer of the normal arrester varistor is Al. On the contrary, a large number of O elements appear in the aluminum electrode layer of the defective and cracked arrester valve varistors. The main element in the side insulation layer of the normal arrester valve varistor is C, while the content of O in the side insulation layer of the defective and burst arrester valve

varistors significantly increases. The insulating layer on the side of the burst arrester has broken down. Therefore, the Zn element inside the varistor is detected on the insulating layer. According to Figure 9c,e and Figure 8c,e, the film composition on the surface of the aluminum layer is alumina. There may be two reasons for the size difference between the alumina film and the film gap of the defective varistor and the exploded varistor: (1) With the development of the oxidation reaction, small films are connected into larger films; (2) When the exploded varistor flows through a large current, it produces a large amount of heat, and the thermal stress may increase the gap between the alumina films.

Based on the results obtained by the SEM and EDS tests, it can be deduced that the main reason for the internal arrester defects is the moisture. The O element enters the arrester in the form of water vapor, while a chemical reaction occurs in the aluminum electrode layer and side insulation layer of the varistor. Consequently, an oxide film appears at the edge of the aluminum electrode layer and the insulation effect of the side insulation layer is reduced. The existence of the oxide film leads to the electric field distortion at the edge of the aluminum electrode layer of the varistor, which easily results in a partial discharge. Simultaneously, due to the deterioration of the side insulating layer, a discharge may develop along the insulating layer, resulting in the surface flashover of the varistor. Finally, the arrester may have thermal collapse under the combined action of external overvoltage and internal insulation defects.

*4.3. Symptoms of Varistor Deterioration Process*

In order to establish the relationship between micro changes and macro symptoms in the process of varistor deterioration, experimental research was carried out. The insulation resistance of varistors in normal, defective, and exploded conditions is tested. An insulation resistance tester is used to test the insulation resistance of the side insulation of varistors. The distance between the test electrodes is 5mm. The test site is shown in Figure 10.

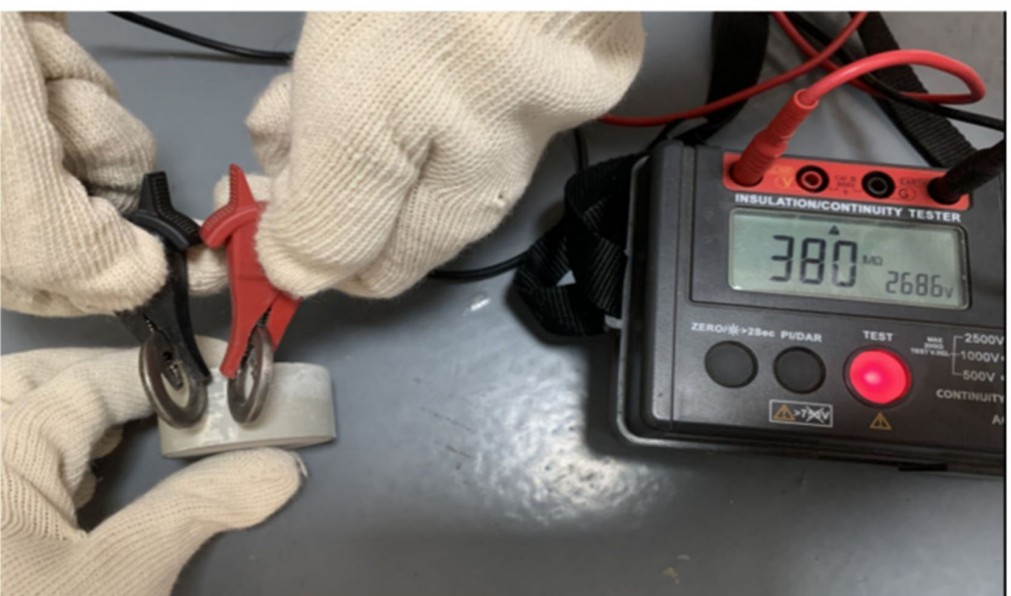

**Figure 10.** Insulation resistance test.

Three test points are selected for the insulation resistance test of the varistor in each state, and the average value of the insulation resistance of each varistor is shown in Figure 11.

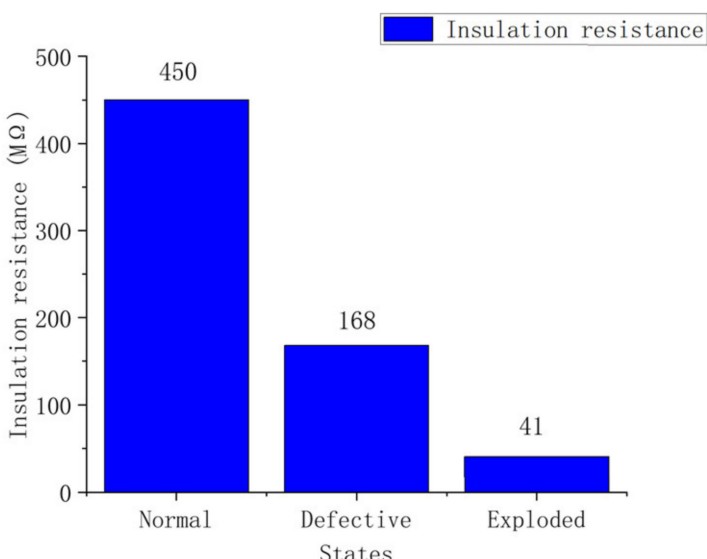

**Figure 11.** Insulation resistance of different states varistors.

With the change of varistor states, the side insulation resistance also changes. Compared with the normal varistor, the insulation resistance of the defective varistor is significantly lower, and the insulation resistance of the exploded varistor is the lowest. Combined with the SEM and EDS test results, it can be seen that there are defects in the sealing performance of the arrester, resulting in damp in the internal varistors. After the varistors are dampened, the aluminum electrode layer and side insulation layer are chemically reflected, and the surface morphology and chemical composition are constantly changing. A symptom of varistor moisture is the reduction of insulation resistance. The deeper the varistor deterioration, the lower the insulation resistance. Finally, the varistor with low side insulation resistance is prone to surface flashover under the action of overvoltage, and even cause the explosion failure of the roof arrester.

## 5. Conclusions

The conclusions of the experimental research on the three lightning arrester samples (normal, defective, and cracked) can be summarized as follows:

(1) When the arrester has internal defects, it may reduce the DC reference voltage, increase the leakage current under 0.75 times the DC reference voltage, and increase the resistive current under continuous operation voltage. The defective arrester can be detected by the routine electrical performance test of the arrester.

(2) After the defective arrester varistor is restored to dry, its electrical performance is close to the factory value. The arrester can then be detected by the electrical performance test under the internal damp state. Once it is dry, the electrical performance will return to its normal state, while the defects still exist. Therefore, it is recommended to increase the maintenance frequency of the roof arrester and detect the defective arrester in time.

(3) The aluminum electrode layer and side insulation layer of the defective and cracked arrester varistors changes in micro morphology, while a high content of O element is detected. This indicates that the main reason for the internal arrester defect is moisture. The technical requirement of improving the sealing performance of the EMU roof arrester and surge current testing including 10 kA 4/10 μs during routine test are suggested.

(4) Not only the roof arrester, but also the arresters installed at various positions may be affected by moisture, resulting in hidden dangers. In order to improve the reliability of the arrester, the following three problems need to be further studied: (a) improve the sealing performance of the arrester; (b) improve the manufacturing process of varistors to improve the chemical stability of aluminum electrode and side insulating material; (c) develop on-line diagnosis technology for the mall defects of arresters.

**Author Contributions:** Q.Z. designed the analysis, wrote the original and revised the manuscript, and conducted data analysis and details of the work. S.W. designed the research experiment. X.D. and M.L. collected the data and conducted the analysis. Q.O. and F.L. designed the research experiment and guided the direction of the work. All authors have read and agreed to the published version of the manuscript.

**Funding:** This research received no external funding.

**Data Availability Statement:** The data that support the findings of this study are available from the corresponding author upon reasonable request.

**Acknowledgments:** CRRC Changchun Railway Vehicles Co., Ltd., provided support in the arrester voltage measurement.

**Conflicts of Interest:** The authors have no conflict to disclose.

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
