# Peer review of "Research of ZnO Arrester Deterioration Mechanism Based on Electrical Performance and Micro Material Test"

_electronics, doi:10.3390/electronics10212624_

Round 1

Reviewer 1 Report

In this paper, the author presented the electrical performance test and material micro-test of roof arresters in three states to study the internal causes of roof arrester explosion and its deterioration mechanism. Their results indicate that during the long-term operation, the varistor may damp and the aluminum electrode layer and side insulation layer of the varistor may deteriorate. This deterioration increases the content of the O element and multiple film structures are formed on the surface. The authors suggested an Improved sealing performance and optimized maintenance process. 

Generally, the paper is well written and acceptable after revision.

  1. Missing the scale bar. It will be good to include a scale bar to Figures 1, 4,5, and 6
  2. In the SEM comparison, the author should indicate the reason behind the increased crack gap in the defective and exploded varistors. Could it be the high temperature that affected the surface? Possibly the thermal stress occurred in the interior of the varistor that caused the burst damage.

Author Response

Dear expert,

Thank you for your valuable comments. For each comment, the manuscript has been modified. Please find the modification in attached PDF file below.

Best wishes,

Qizhe ZHANG

Reviewer 2 Report

Deterioration of roof arresters is a progressive process in nature. Hence, the study would be more complete if the focus is shifted to how the deterioration symptoms connect the different states/phases considered in this work.

The conclusion does not give any sense of any relatively new knowledge of deterioration mechanism being produced.

Author Response

Dear expert,

Thank you for your valuable comments. For each comment, the manuscript has been modified. Please find the modification in detials attached in the PDF file below.

Best wishes,

Qizhe ZHANG

Reviewer 3 Report

The manuscript provides experimental evidence of the degradation effects of varistor-based surge arresters used in traction systems; this work may have significant engineering implications on demanding applications of surge protection industry not directly related to this study, such as [R0] covered also by electronics/MDPI. Although the work is interesting some clarifications and discussion points should be covered in a revised version of the manuscript.

[R0] A. Formisano et al., “Modeling of PV module and DC/DC converter assembly for the analysis of induced transient response due to nearby lightning strike,” Electronics, vol. 10, no. 2:120, 2021.

  1. Introduction: Since the general framework of surge arrester challenges and applications is provided in the introduction the literature review should be enriched with recent books and working group reports reflecting the state of the art in the field associated with varistor material [R1, R2], varistor failure and inhomogeneity issues [R3, R4] and electrothermal behavior [R5].

[R1] F. Greuter, ZnO varistors: From grain boundaries to power applications, in book “Oxide Electronics,” Ed. A. Ray, Chapter 8, 2021.

[R2] J. He, Metal Oxide Varistors: From Micorstructure to Macro-Characteristics, Tsinghua University Press, 2019.

[R3] Z. Topcagic et al., “Varistor Electrical Properties: Microstructural Effects,” Encyclopedia of Materials: Technical Ceramics Glasses, Ed. M. Pomeroy, Elsevier, vol. 3, pp. 254-271, Mar. 2021.

[R4] CIGRE Working Group A3.25, “MO Surge Arresters – Metal Oxide Resistors and Surge Arresters for Emerging System Conditions,” Technical Brochure 696, Aug. 2017.

[R5] Y. Späck-Leigsnering, Electrothermal modeling, simulation and optimization of surge arresters, PhD Thesis, Darmstadt University, 2019.

The use of parallel clearance, such as the one presented in [24] of the manuscript, for energy coordination of surge arresters may have significant drawbacks, such as follow current from the grid which can be fatal in the case of DC applications; which is the continuous operating voltage of the surge arrester in the application of the paper?

  1. Test object and Research Method: Please add in the list of the technical specification of the surge arrester as 6th bullet the capacitance of the surge arrester; this may be important for AC conduction, see for example [R6].

[R6] S. Blatt at al., “Mathematical model for numerical simulation of current density in microvaristor filled insulation materials,” IEEE Trans. on Dielectrics and Electrical Insulation, vol. 22, no. 2, 2015.

Please add in the list of the technical specification of the surge arrester as 7th bullet the energy absorption capability (kJ/kV) or/and Class of the surge arrester; this may be important for the failure probability determination.

Please briefly explain the meaning and function of the pantograph?

Table 1: I suggest replacing the symbol I0.75U1mA with [email protected] for the shake of the readers.

Figs 2 and 3. Provide technical details for instruments, especially for the current measurement technique which is critical on the accuracy of the experimental results in more demanding measurements such as those of [R7]. Also, a picture of the experimental arrangement would be helpful for the readers. Differences between the factory test and the “normal” surge arrester may be attributed to different measurement techniques and/or ambient temperature. Please update Figs 2 and 3 accordingly in the revised manuscript.

[R7] S. Passon et al., “Metrology for very fast current transients,” in Proc. Int. Conf. High Voltage Eng. Applic., Athens, Greece, 2018.

  1. Macroscopic characteristics

Table 2: Which is the time interval for capturing the current measurement. For DC application there is a case of time-varying current conduction; for example, records are taken 100 ms after voltage application in [R8] and in international standards, such as IEC 61643-11, DC conduction is commonly captured in seconds range; this may affect also results in Table 3. Please clarify in the revised manuscript.

[R8] E. V. Staikos et al., “Low-frequency response of low-voltage metal-oxide varistors used for telecommunication systems protection,” in Proc. 21st Int. Conf. High Voltage Eng. Applic., China, 2020.

Table 2: Did you observe any polarity effect on the DC current flow of the defective surge arrester? See for example [R9]. Please clarify and comment.

[R9] X. Zhao et al., “DC Ageing mechanism of Co2O3-dopes ZnO varistors,” Energies, 2021.

Fig 5: There is a dot in Fig5a that maybe indicate bad contact and high roughness of the varistor block surface; this is probably associated with findings of paragraph 4.1; please comment.

Fig 6: (i) Designate the HV side of the surge arrester in Fig6a for the shake of the readers. (ii) This failure mode is not safe; please provide in the text which is the disconnection mechanism of a failed surge arrester from the grid (external or internal). There are several patents and practices in the surge protection industry that involve explosive disconnector [R10] and/or thermal disconnection mechanisms [R11], wired [R12] or wireless [R13] smart detection & disconnection systems, oil insulation of varistor blocks [R4] or even a metal enclosure of arresters (commonly found in military applications) so as to ensure safe failure mode and prevent an explosion. Provide a relative discussion in the revised manuscript.

[R10] V. Hinrichsen, Metal-Oxide Surge Arresters in High-Voltage Power Systems Fundamentals, 3rd edition, Siemens AG, Berlin and Darmstadt, 2012.

[R11] J. Vrhunc et al., “Surge protective device modules including integral thermal disconnect and methods including same,” United States Patent, Patent No. US 10,340,110 B2, Jul. 2, 2019.

[R12] T. E. Tsovilis et al., “DC overload behavior of low-voltage varistor-based surge protective devices,” IEEE Trans. Power Delivery, vol. 35, no. 5, pp. 2541-2543, Oct. 2020.

[R13] J. Kristainsson, Line surge arresters - contribution to power quality: line side protection, Presentation to ABB Africa Channel Partner Event, 2016.

[R4] CIGRE Working Group A3.25, “MO Surge Arresters – Metal Oxide Resistors and Surge Arresters for Emerging System Conditions,” Technical Brochure 696, Aug. 2017.

Table 3: It would be interesting to give a hint on the impedance of the varistor blocks of the exploded surge arrester or briefly explain in the text why you did not perform any measurements to them.

Table 4: provide the thickness of varistor blocks and the residual voltage at 100 kA so as to be able to evaluate the electric field stress (residual voltage peak / thickness) that the side insulation breaks down at 4/10μs test.

  1. SEM and EDS of varistors

The main conclusion of this study is important and summarized in the following sentence “the main reason of the internal arrester defect is moisture” (line 285). This should be also clearly written in slightly revised conclusions and maybe calls for surge current testing, including 100 kA 4/10 μs, that includes a climatic chamber as recently proposed in [R14]. This conclusion may affect international committees on revising the surge testing requirements for surge arresters and surge protective devices.

[R14] T. E. Tsovilis, “Critical insight into performance requirements and test methods for surge protective devices

-------------

Note: There are some typos in the manuscript that undermine the value of the work – for example, the word “port” instead of “part” in line 125, “..” instead of “.” in line 233, “μS” instead of “μs” in line 233. Scan the full text for typos including references.

Author Response

(The authors gave the same response as above.)

Round 2

Reviewer 2 Report

N/A

Reviewer 3 Report

ΟΚ - the revised manuscript is satisfactory.